# The cryo-EM structure of a 12-subunit variant of RNA polymerase I reveals dissociation of the A49-A34.5 heterodimer and rearrangement of subunit A12.2

Lucas Tafur[1,2], Yashar Sadian[1], Jonas Hanske[1], Rene Wetzel[1], Felix Weis[1], Christoph W Müller[1]*

[1]Structural and Computational Biology Unit, European Molecular Biology Laboratory, Heidelberg, Germany; [2]Collaboration for joint PhD degree, European Molecular Biology Laboratory and Heidelberg University, Faculty of Biosciences, Heidelberg, Germany

**Abstract** RNA polymerase (Pol) I is a 14-subunit enzyme that solely transcribes pre-ribosomal RNA. Cryo-electron microscopy (EM) structures of Pol I initiation and elongation complexes have given first insights into the molecular mechanisms of Pol I transcription. Here, we present cryo-EM structures of yeast Pol I elongation complexes (ECs) bound to the nucleotide analog GMPCPP at 3.2 to 3.4 Å resolution that provide additional insight into the functional interplay between the Pol I-specific transcription-like factors A49-A34.5 and A12.2. Strikingly, most of the nucleotide-bound ECs lack the A49-A34.5 heterodimer and adopt a Pol II-like conformation, in which the A12.2 C-terminal domain is bound in a previously unobserved position at the A135 surface. Our structural and biochemical data suggest a mechanism where reversible binding of the A49-A34.5 heterodimer could contribute to the regulation of Pol I transcription initiation and elongation.
DOI: https://doi.org/10.7554/eLife.43204.001

*For correspondence:
christoph.mueller@embl.de

Competing interests: The authors declare that no competing interests exist.

## Introduction

RNA polymerase I (Pol I) is a eukaryotic, 14-subunit enzyme that solely transcribes pre-ribosomal (rRNA) from ribosomal DNA (rDNA) repeats. Although all three eukaryotic RNA polymerases (Pol I, Pol II and Pol III) share a structurally conserved 10-subunit core and a 2-subunit stalk, they have evolved distinct structural features, including accessory subunits, and rely each on a unique specialized set of general transcription factors (*Engel et al., 2018*; *Khatter et al., 2017*; *Vannini and Cramer, 2012*). Before the first structures became available, functional studies already suggested that Pol I had adapted to accommodate the transcriptional needs of ribosome production resulting in differences in its regulation, initiation and elongation compared to the well-studied Pol II system to promote fast initiation and processivity (*Albert et al., 2012*). Accordingly, Pol I relies on a simpler transcription initiation machinery compared to Pol II (*Keener et al., 1998*), and similar to Pol III, has incorporated Pol II transcription factor-like subunits during evolution (*Engel et al., 2018*; *Khatter et al., 2017*; *Vannini and Cramer, 2012*).

In recent years, structural information available for *Saccharomyces cerevisiae* (yeast) Pol I has increased dramatically, revealing the structural basis of the Pol I-specific functional adaptations. In the first crystal structures, Pol I formed a dimer, thereby locking the enzyme in an inactive conformation (*Engel et al., 2013*; *Fernández-Tornero et al., 2013*). Although the core structure was overall

conserved compared to Pol II, the Pol I DNA-binding cleft was very wide, and it was occupied by the C-terminal domain of Pol I-specific subunit A12.2 and a DNA-mimicking loop/expander element that occupies the position of the DNA-RNA transcription bubble. This wide cleft conformation resulted in the unfolding of the bridge helix, a conserved element that connects the two biggest subunits in multi-subunit RNA polymerases and plays an important role during catalysis (*Weinzierl, 2011*). Subsequent cryo-electron microscopy (cryo-EM) structures of the Pol I elongation complex (EC) revealed that binding to a DNA-RNA scaffold promoted the closure of the DNA-binding cleft, thus freeing the active site for nucleotide binding and causing the folding of the bridge helix (*Neyer et al., 2016*; *Tafur et al., 2016*). The EC structures were very similar to those observed for Pol II (*Kettenberger et al., 2004*) and Pol III (*Hoffmann et al., 2015*), highlighting that the actively transcribing Pol I adopts a conserved conformation and suggesting that the enzymatic mechanism of nucleotide addition is functionally conserved. However, mutating specific conserved residues in the Pol I and Pol II active sites appear to have different effects in vitro (*Viktorovskaya et al., 2013*), suggesting that other elongation intermediates might reveal previously uncharacterized differences between Pol I and Pol II.

Structural data of the basal Pol I initiation complex have revealed a very different, much simpler architecture compared to Pol II (*Engel et al., 2017*; *Han et al., 2017*; *Sadian et al., 2017*). This simplification is further supported by the incorporation of transcription factor-like functions into Pol I subunits: In Pol I, the A49-A34.5 heterodimer (hereafter referred also as 'heterodimer') has been proposed to function as both, a TFIIF- and TFIIE-like factor, participating during transcription initiation and elongation (*Geiger et al., 2010*; *Vannini and Cramer, 2012*). A49 has two domains connected by a linker, each of which appears to have evolved functionally distinct properties. While the A49 C-terminal tandem winged helix domain (tWH) has structural homology to TFIIE, the N-terminal A49 domain forms a dimer with the A34.5 subunit which adopts a triple β-barrel structure that resembles the Rap74/30 module of TFIIF (*Geiger et al., 2010*). The heterodimer is anchored to the core enzyme by interactions through the A49-A34.5 dimerization domain and by an extended surface between the long C-terminal tail of A34.5 (A34.5-Ct) and Pol I's second biggest subunit A135 (*Engel et al., 2013*; *Fernández-Tornero et al., 2013*). However, as the dimerization domains contribute most to the binding, deletion of either subunit results in a Pol I enzyme lacking both subunits (*Gadal et al., 1997*; *Pilsl et al., 2016*).

Since its discovery, Pol I has been shown to exist in two different conformations that differ by the presence of the heterodimer, which can be reversibly dissociated (*Huet et al., 1975*). The form lacking A49-A34.5, termed Pol I*, has reduced transcriptional specificity and activity compared to the complete Pol I enzyme (*Huet et al., 1976*). Although the heterodimer appears to increase the processivity of Pol I, details of its function are still unknown. Neither A49 (*Liljelund et al., 1992*) nor A34.5 (*Gadal et al., 1997*) are essential genes, and Pol I* has been proposed to co-exist with Pol I in vivo (*Gadal et al., 1997*). Deletion of topoisomerase I causes a very strong growth defect in yeast only when combined with a deletion of A34.5 (*Gadal et al., 1997*) suggesting that A34.5 is important for relieving topological stress during rDNA transcription. In vitro, the A49-A34.5 heterodimer has a stronger effect on promoter-dependent transcription than on non-specific transcription, while addition of the A49 tWH domain is sufficient to restore promoter-dependent and non-specific transcription (*Pilsl et al., 2016*). Overall, the data suggest that the heterodimer is functionally important for transcription initiation and/or (early) elongation. However, the functional and physiological relevance of Pol I* has not been elucidated to date. Furthermore, it is not clear if the heterodimer participates in all phases of transcription, or only during initiation and early elongation.

The Pol I-specific subunit A12.2 also contains additional built-in functionality. A12.2 shares homology with Pol II subunit Rpb9 in its N-terminal domain and the Pol II cleavage factor TFIIS in its C-terminal domain (*Ruan et al., 2011*). While the role of TFIIS in RNA cleavage is well established (*Cheung and Cramer, 2011*), Rpb9 appears to regulate transcription elongation (*Hemming et al., 2000*), proofreading (*Knippa and Peterson, 2013*) and transcription-coupled DNA repair (*Li et al., 2006*). The A12.2 C-terminal Zn ribbon domain (A12.2C) is required for the Pol I intrinsic RNA cleavage activity (*Kuhn et al., 2007*) and adopts a similar position as TFIIS in the cleft in unbound (apo) Pol I (*Engel et al., 2013*; *Fernández-Tornero et al., 2013*; *Neyer et al., 2016*) as well as in Pol I bound only to DNA (*Sadian et al., 2017*; *Tafur et al., 2016*), but is excluded from the active site upon formation of the EC (*Neyer et al., 2016*; *Tafur et al., 2016*). The exact position of A12.2C, however, has not been determined in the context of an actively transcribing complex. While deletion

of A12.2C does not cause any growth defect, deletion of the A12.2 N-terminal Zn ribbon domain (A12.2N) produces a similar effect as deletion of the complete protein (*Van Mullem et al., 2002*). Interestingly, deletion of either the complete A12.2 or A12.2N also alters the nucleolar localization of Pol I, suggesting that A12.2 is important for Pol I integrity.

Studies to date suggest a functional interplay between the Pol I A49-A34.5 heterodimer and subunit A12.2. The heterodimer stimulates A12.2-mediated RNA cleavage in vitro (*Geiger et al., 2010*), the latter which is important for Pol I backtrack recovery (*Lisica et al., 2016*). A12.2N interacts directly with the dimerization domain of A49, thus stabilizing the anchoring of the heterodimer (*Engel et al., 2013*; *Fernández-Tornero et al., 2013*). Recently, A12.2 has also been proposed to be important for transcription initiation *in vivo* and *in vitro*, especially in the absence of A49 (*Darrière et al., 2018*). Combined with the reduced number of general transcription factors required for productive transcription initiation, the Pol I A49-A34.5 heterodimer and subunit A12.2 might promote the high initiation rate observed on rDNA repeats (*French et al., 2003*).

In this work, we describe the cryo-EM structures of Pol I and spontaneously formed Pol I* bound to a DNA-RNA scaffold and the nucleotide analog GMPCPP. These structures reveal a previously unobserved relationship between A12.2 and A34.5, provide the structural basis for the exclusion of the heterodimer from the core enzyme, and suggest mutually exclusive binding of the A49-A34.5 heterodimer and A12.2C during the Pol I transcription cycle.

## Results

### Cryo-EM structures of the GMPCPP-bound Pol I elongation complexes (EC)

In order to better understand the catalytic mechanism of Pol I, we incubated the Pol I EC with the non-hydrolysable nucleotide analog GMPCPP as previously used for Pol II (*Kettenberger et al., 2004*; *Wang et al., 2006*). The Pol I EC was prepared as previously described (*Tafur et al., 2016*) except that 1 mM $MgCl_2$ was included in the buffer (Materials and methods). 5768 micrograph movies were collected on a FEI Titan Krios equipped with a K2 direct electron detector, and processed with RELION 2.0 (*Kimanius et al., 2016*). After sorting particles with 2D and 3D classification, an unexplained extra density next to the A135 surface was observed in most of the particles with a closed cleft and strong DNA-RNA density, concomitant with streaky and weak density for the A49-A34.5 heterodimer. To better resolve this density, particles were classified using a mask in this region (*Figure 1—figure supplement 1*). This revealed that the extra density corresponded to A12.2C (*Figure 1*). In total, 63% of all particles selected after the first unmasked 3D classification step did not have the heterodimer bound and showed density for A12.2C in this new position (named Pol I* in analogy to RNA polymerase A* (*Huet et al., 1975*)), while only 37% represented the 14-subunit Pol I. Extensive 3D classification ultimately yielded two different nucleotide-bound ECs: 12-subunit Pol I* EC lacking the heterodimer, which was refined to 3.18 Å resolution, and 14-subunit Pol I EC, which was refined to 3.42 Å resolution (*Figure 1—figure supplement 2*). The overall conformation of both Pol I forms is very similar, with the exception of the presence/absence of the heterodimer, the previously unobserved position of A12.2C and a slight difference in the conformation of the clamp, and resemble previously published structures (*Figure 1*) (*Neyer et al., 2016*; *Tafur et al., 2016*). Interestingly, an apo Pol I* reconstruction at 3.21 Å resolution was also obtained with a similar conformation as previously observed for the cryo-EM structures of monomeric Pol I (*Neyer et al., 2016*; *Pilsl et al., 2016*), highlighting that the presence of the heterodimer and the novel position of A12.2C do not impose any conformational constraints on the Pol I core (*Figure 1—figure supplement 3*). Models were built using previous Pol I structures as a starting point and were real-space refined, yielding structures with excellent stereochemistry (*Table 1*).

### The A12.2 C-terminal domain alternates between a TFIIS-like and an Rpb9-like position

In the 12-subunit Pol I* EC, lacking the heterodimer, the A12.2C occupies a novel position next to the A135 surface (*Figure 1A*). This new position overlaps with the A34.5-Ct in the complete, 14-subunit Pol I EC (*Figure 1B*), where A12.2C is disordered and only density up to residue 67 is observed (*Neyer et al., 2016*; *Tafur et al., 2016*). The new position of A12.2C resembles that of the

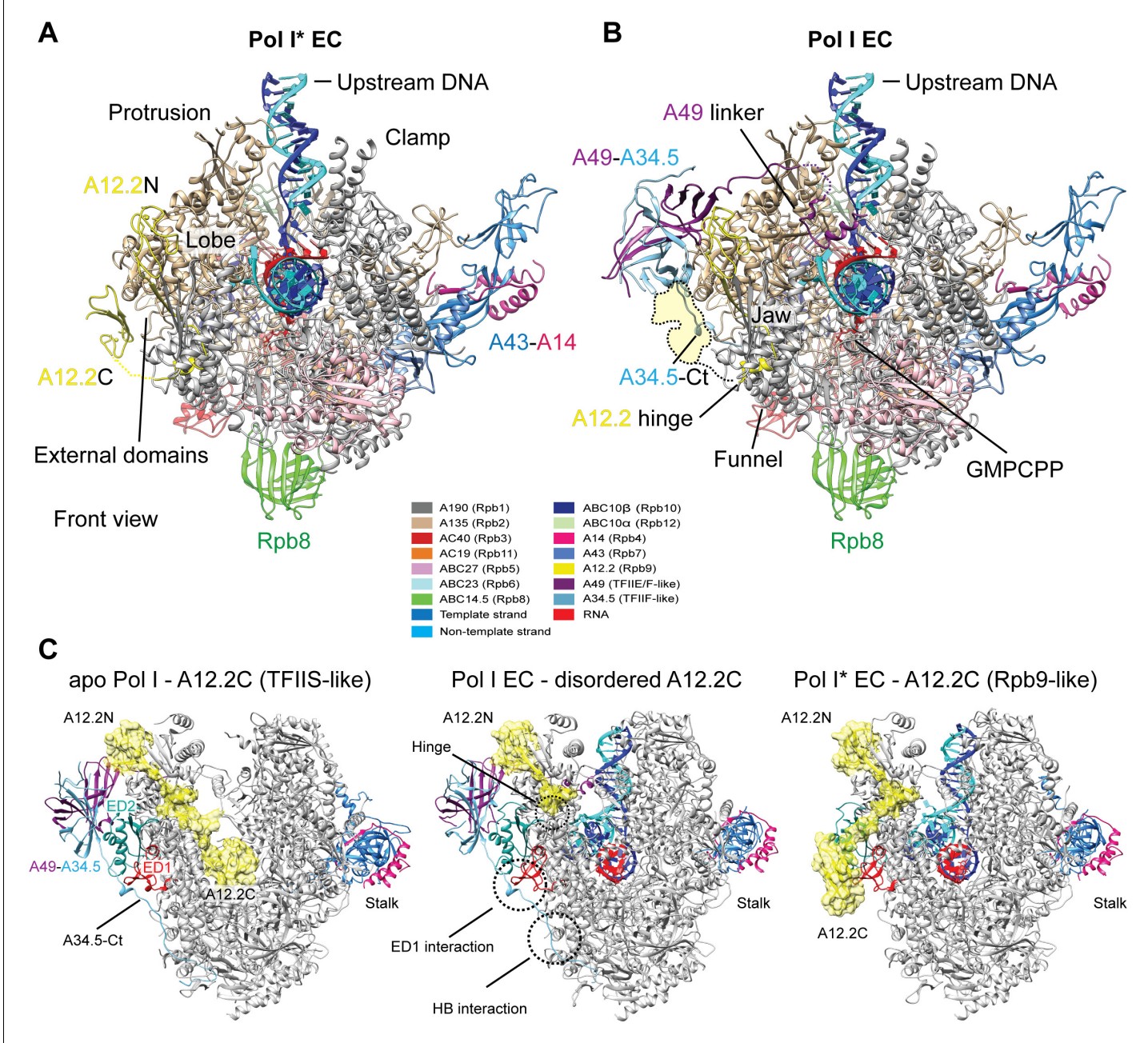

**Figure 1.** Structures of the Pol I* EC and Pol I EC bound to GMPCPP. (**A**) In the Pol I* EC, the A49-A34.5 is absent and the A12.2C adopts a position on the A135 surface. The overlap between this position and the C-terminal domain of A34.5 is indicated in the Pol I EC as a dashed yellow surface. (**B**) The 14-subunit Pol I bound to a DNA-RNA scaffold is shown colored according to the subunits indicated in the legend. In this conformation, only up to residue 67 of A12.2 is observed (A12.2 hinge), while the C-terminal domain (A12.2C) is disordered. (**C**) Comparison between the apo (left), Pol I EC (middle) and Pol I* EC (right) reveals that the A12.2C can alternate between TFIIS-like (apo) or Rpb9-like (right) positions. Movement of the A12.2C is around a hinge at residue 67, also indicated in the Pol I EC (**A**). The position of the External domain 1 (ED1) and hybrid binding (HB) interaction surfaces are indicated in the Pol I EC. A12.2 is shown as ribbon diagram and yellow surface (not EM density) for easier visualization. See also *Figure 1—figure supplements 1–3*.

DOI: https://doi.org/10.7554/eLife.43204.002

The following figure supplements are available for figure 1:

**Figure supplement 1.** Cryo-EM data and processing.

DOI: https://doi.org/10.7554/eLife.43204.003

**Figure supplement 2.** Average and local resolution estimates for the reconstructions.

DOI: https://doi.org/10.7554/eLife.43204.004

*Figure 1 continued on next page*

*Figure 1 continued*

**Figure supplement 3.** Structure of the apo Pol I*.
DOI: https://doi.org/10.7554/eLife.43204.005

C-terminal domain of Pol II subunit Rpb9, outside of the DNA-binding cleft (*Figure 1C*, right), and is distinct from the previously reported TFIIS-like position near the active site (*Figure 1C*, left). The A12.2C can move between these positions by rotating around a hinge located at residues 66–67 (*Figure 1A*, *Figure 1C*, middle). Whereas binding to the TFIIS-like position is only possible when the DNA-binding cleft is open (*Engel et al., 2013*; *Fernández-Tornero et al., 2013*) or partially open (*Neyer et al., 2016*; *Sadian et al., 2017*; *Tafur et al., 2016*), binding to the Rpb9-like position can only occur when the heterodimer is absent.

Detailed analysis of Pol I* and Pol I reveals that different interactions occur in two areas of the A135 surface (*Figure 2A*, *Figure 1C*, middle). The first area involves part of the A135 External Domain 1 (ED1), which interacts either with the A34.5-Ct (in Pol I) or A12.2C (in Pol I*). The second area corresponds to part of the A135 Hybrid Binding (HB) domain (residues 989 to 1000), which in Pol I interacts with the A34.5-Ct but in Pol I* interacts with the A135 N-terminal tail (A135-Nt), which folds back towards the HB domain (*Figure 2A*). The A135-Nt effectively acts as a switch, changing its positioning to allow or to prevent A34.5-Ct binding to the HB domain. Both A12.2C and A34.5-Ct form similar interactions with the Pol I core, as both interact with two neighboring asparagine residues in the A135 ED1 (N683 and N684) (*Figure 2B,C*) and an aspartate residue (D990) in the A135 HB domain (*Figure 2D,E*).

In the monomeric apo Pol I, A12.2C can still occupy the TFIIS-like position (*Neyer et al., 2016*). However, in the apo Pol I*, despite being sufficient space for accommodating A12.2C in the TFIIS-like position, A12.2C is observed in the Rpb9-like position (*Figure 1—figure supplement 3*). The presence of the heterodimer in the enzyme could thus promote binding of A12.2C to the TFIIS-like

**Table 1.** Data collection and refinement statistics.

| | Pol I (core) EC + GMPCPP | Pol I EC + GMPCPP | Pol I* EC + GMPCPP | Apo Pol I* |
|---|---|---|---|---|
| Data collection | | | | |
| Particle number | 54,017 | 30,232 | 182,488 | 73,660 |
| Pixel size (Å/pix) | 1.04 | 1.04 | 1.04 | 1.04 |
| Average resolution (Å) | 3.18 | 3.42 | 3.18 | 3.21 |
| B-factor | −44.5 | −34.2 | −92.9 | −99.6 |
| EMDB code | EMD-0240 | EMD-0238 | EMD-0239 | EMD-0241 |
| Refinement statistics* | | | | |
| PDB code | 6HLR | 6HKO | 6HLQ | 6HLS |
| CC (atoms)[†] | 0.816 | 0.804 | 0.796 | 0.797 |
| RMSD (bonds) | 0.007 | 0.006 | 0.006 | 0.007 |
| RMSD (angles) | 1.22 | 1.18 | 1.18 | 1.25 |
| Clashscore | 4.74 | 5.27 | 5.13 | 5.17 |
| Rotamer outliers (%) | 0.12 | 0.14 | 0.09 | 0.32 |
| C-beta deviations (%) | 0 | 0 | 0 | 0 |
| Ramachandran plot | | | | |
| Outliers (%) | 0 | 0 | 0 | 0 |
| Allowed (%) | 4.9 | 5.64 | 4.59 | 5.48 |
| Favored (%) | 95.1 | 94.36 | 95.41 | 94.61 |
| Molprobity score | 1.58 | 1.67 | 1.59 | 1.65 |

*Calculated with Molprobity.
†From PHENIX real space refinement.
DOI: https://doi.org/10.7554/eLife.43204.006

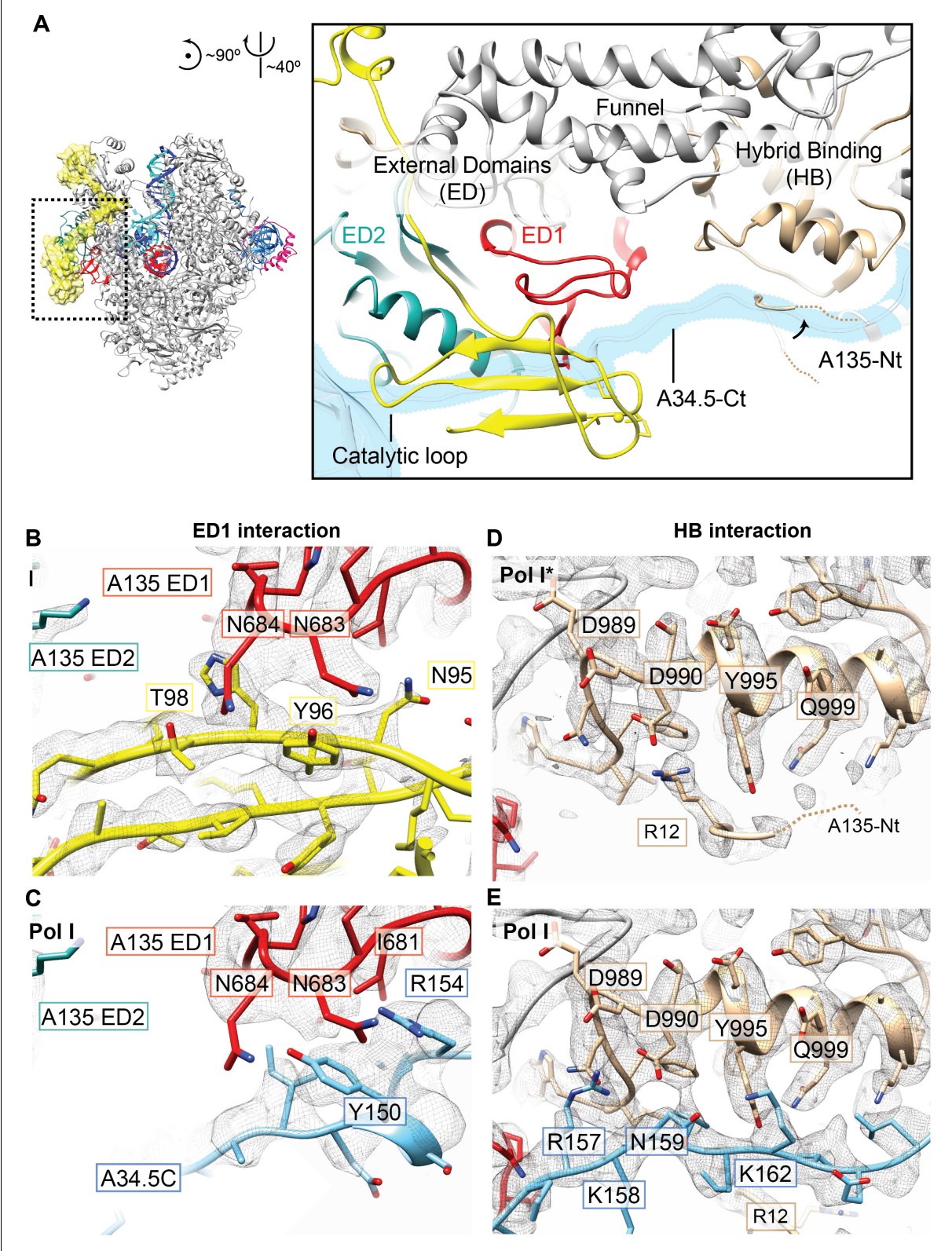

**Figure 2.** Interactions of the A12.2C with the A135 External domain one and Hybrid binding domain. (A) Two interfaces are differently arranged in Pol I* versus Pol I. Both A34.5-Ct and A12.2C can bind to the A135 External Domain 1 (ED1, red), and A34.5-Ct and the N-terminal tail of A135 (A135-Nt) can bind to the A135 Hybrid Binding domain (HB). (B). In the ED1, the A12.2C interacts with both A135 N683 and N684 through Y96 and T98, respectively. (C). In the ED1, the A34.5-Ct interacts with A135 N683 and N684 through R154 and Y150, respectively. (D) In the HB surface, the A135-Nt

*Figure 2 continued on next page*

*Figure 2 continued*
folds back and positions R12 next to D990. (**E**) A34.5-Ct interacts with D990 from the HB domain through R157. Densities shown for panels B-E are from the sharpened Pol I* and Pol I EC (+GMPCPP). See also *Figure 1—figure supplement 2*.
DOI: https://doi.org/10.7554/eLife.43204.007

site (when accessible) by blocking the Rpb9-like binding site. In apo Pol I*, the change in the position of A12.2C also shifts the A12.2N by ~3 Å towards the jaw, and part of the latter appears to move towards the A12.2 linker, likely to stabilize its position (*Figure 1—figure supplement 3*). Interestingly, both domains move relative to a region of A12.2 (residues ~ 43–66), which fixes this subunit to the Pol I core. Therefore, the movement of both, the A12.2N and the jaw, accommodate the change in the position of A12.2C.

## A12.2C does not displace A49-A34.5 from the Pol I core

At present, it is unclear why most of the particles lack the heterodimer compared to previous Pol I EC structures (*Neyer et al., 2016*; *Tafur et al., 2016*). It is possible that differences in sample preparation conditions such as changes in the buffer conditions during freezing or the use of a thin layer of carbon in the cryo-EM grids account for the difference. While the cryo-EM structures show that A34.5-Ct and A12.2C compete for the same binding sites in A135, they don't allow to distinguish if A12.2C displaces the heterodimer from the Pol I core or if A12.2C binds only once the heterodimer has dissociated from the enzyme. To test these hypotheses, we performed a series of fluorescence anisotropy experiments, using recombinant heterodimer, where a cysteine has been introduced in the A49 linker region for labelling with Alexa Fluor 594, and endogenously purified Pol I* (*Pilsl et al., 2016*) incubated with DNA (Pol I * EC) (*Figure 3A*). Because the fluorescent signal was low, we performed the experiments with a heterodimer concentration of 100 nM. Compared to the heterodimer alone, we observed an increase in anisotropy in a concentration-dependent manner as we added Pol I* EC (*Figure 3B*). The same experiment using wild type Pol I EC gave a right-shifted curve, indicating an exchange between endogenous heterodimer on wild type Pol I and labelled heterodimer. These data suggest that heterodimer binding to Pol I is reversible, and that A12.2C binding A135 as observed in Pol I* does not irreversibly prevent heterodimer binding. Because a 1:1 binding model did not allow fitting the data, no attempt was made to introduce more complex binding models. Incubation of the Pol I*/A49-A34.5 sample with recombinant A12.2C (residues 79 to 125) for 30 min did not reduce the anisotropy (indicating the release of the heterodimer from Pol I) even at 50-fold molar excess (*Figure 3C*). Although the affinity of A12.2C for the ED1 might further increase when it is constitutively anchored to Pol I by A12.2N. Similarly, incubation of the complex in the presence of GMPCPP did not change the anisotropy of the bound complex even at 20 mM (*Figure 3C*). These results suggest that binding of the A12.2C to the Rpb9-like position is only possible after the heterodimer has dissociated from Pol I.

## ED1 determines binding of the C-terminal domain of the Rpb9-like subunit

Comparison of Pol I* with Pol II and Pol III reveals that while the External Domain 2 (ED2) appears to be structurally more conserved, the Pol I ED1 diverges from its Pol II and Pol III counterparts, as it is smaller and lacks an extension that overlaps with A12.2C in the Rpb9-like position (*Figure 4A*). In Pol II, the Rpb9 C-terminal domain (Rpb9C) also binds the ED1, although differently than A12.2C due to the presence of an extension in the ED1 (*Figure 4B*). Therefore, the Pol I and Pol II ED1 are specifically tailored to bind A12.2C and Rpb9C, respectively. Interestingly, a similar situation is observed in Pol III (*Figure 4C*). The Pol III ED1, as in Pol II, also has an extension in a region that overlaps with the position of A12.2C, but in addition, binding of the C11 C-terminal domain (C11C, equivalent to Pol I A12.2C and Pol II Rbp9C) in an Rbp9C-like position is precluded by the presence of a helix from subunit C53. Accordingly, the C11C adopts a position far from the Pol III ED1 (*Hoffmann et al., 2015*) that differs from the position of both A12.2C and Rpb9C (*Figure 4C*).

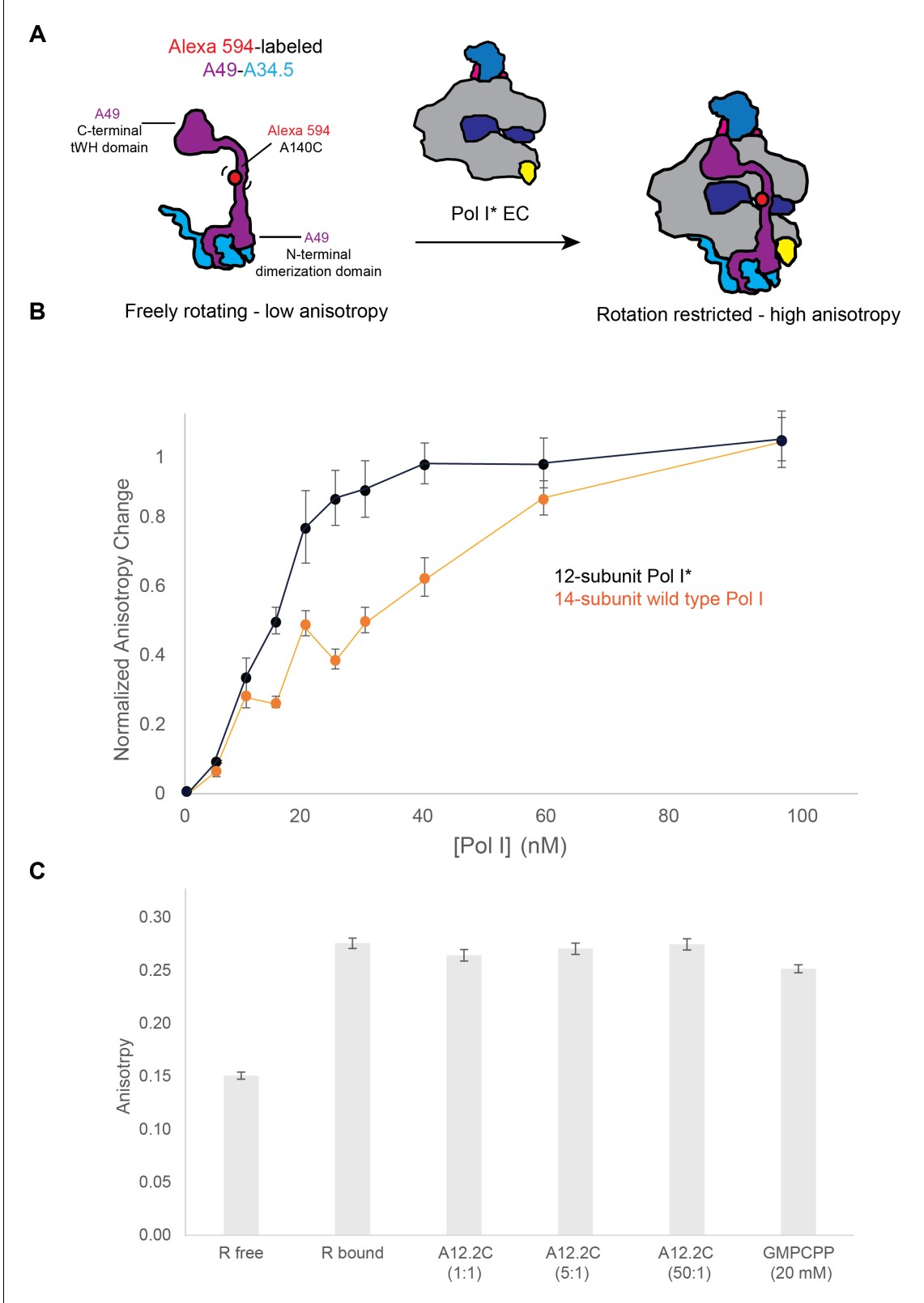

**Figure 3.** Binding of the A49-A34.5 to the Pol I core in vitro. (**A**) Experimental set up. Recombinant A49-A34.5, fluorescently labeled with Alexa 594 at residue 140, was mixed with the reconstituted Pol I* EC. The change in fluorescence anisotropy reflects the binding of A49-A34.5 to the Pol I core (an increase in anisotropy with respect to the free A49-A34.5 represents the formation of the 14-subunit Pol I). (**B**) Experimental data showing the change in fluorescence anisotropy upon binding of fluorescent A49-A34.5 to Pol I* as well as the replacement of endogenous heterodimer in wild type Pol I by

*Figure 3 continued*
fluorescent A49-A34.5. The points shown are an average of three replicates, with the standard deviation. (C) The reconstituted and labeled 14-subunit Pol I EC was incubated with increasing amounts of recombinant A12.2C (residues 70–125) for 30 min. Compared to the Pol I EC, no change in anisotropy is observed at either 1, 5 or 50-fold molar excess of A12.2C or with 20 mM GMPCPP.
DOI: https://doi.org/10.7554/eLife.43204.008

## The active site conformations in Pol I * and Pol I are identical

Because the Pol I* EC reconstruction was obtained in the presence of the non-hydrolysable nucleotide analog GMPCPP and 1 mM $MgCl_2$, we carefully compared the Pol I* active site with the active site in the 14-subunit Pol I reconstruction. As no differences were observed between the active sites, we pooled particles from both EC reconstructions, and classified them by restricting the classification to the core enzyme and the DNA-RNA hybrid using a soft mask and higher weight on the data (*Scheres, 2016*) (*Figure 1—figure supplement 1*, Materials and methods). This strategy also allowed us to resolve two main features from the active site: the binding and interactions of the incoming nucleotide (NTP) substrate (GMPCPP), and the interactions between Pol I and the +1 and +2 bases from the single-stranded non-template strand (NT) (*Figure 5A*).

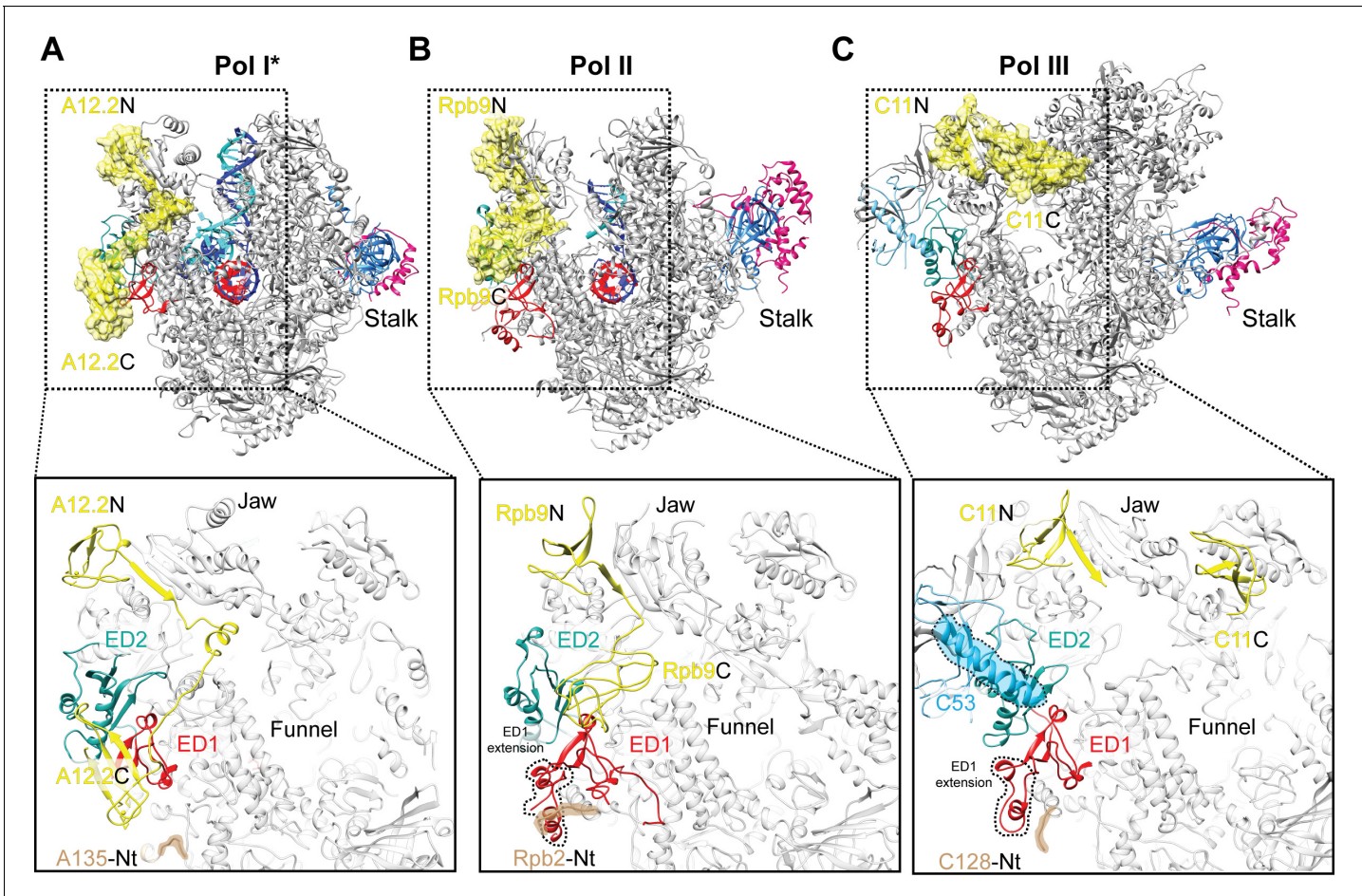

**Figure 4.** Comparison of the positions of the C-terminal domains of Pol I A12.2, Pol II Rbp9 and Pol III C11. The positions of A12.2 (A), Rpb9 (B) or C11 (C) are shown in yellow for Pol I*, Pol II (*Kettenberger et al., 2004*) and Pol III (*Hoffmann et al., 2015*), respectively. While the ED2 is structurally more conserved (light sea green color), the ED1 in Pol II and Pol III are larger than the Pol I ED1 (red). The structure of the ED1 determines the binding mode of Pol I A12.2C and Pol II Rpb9C, while in Pol III the presence of C53 induces a different binding site for C11C far from the ED. The position of the N-terminal tail of the second largest subunit is also indicated for each polymerase, as well as the extension in the ED1 of Pol II and Pol III.
DOI: https://doi.org/10.7554/eLife.43204.009

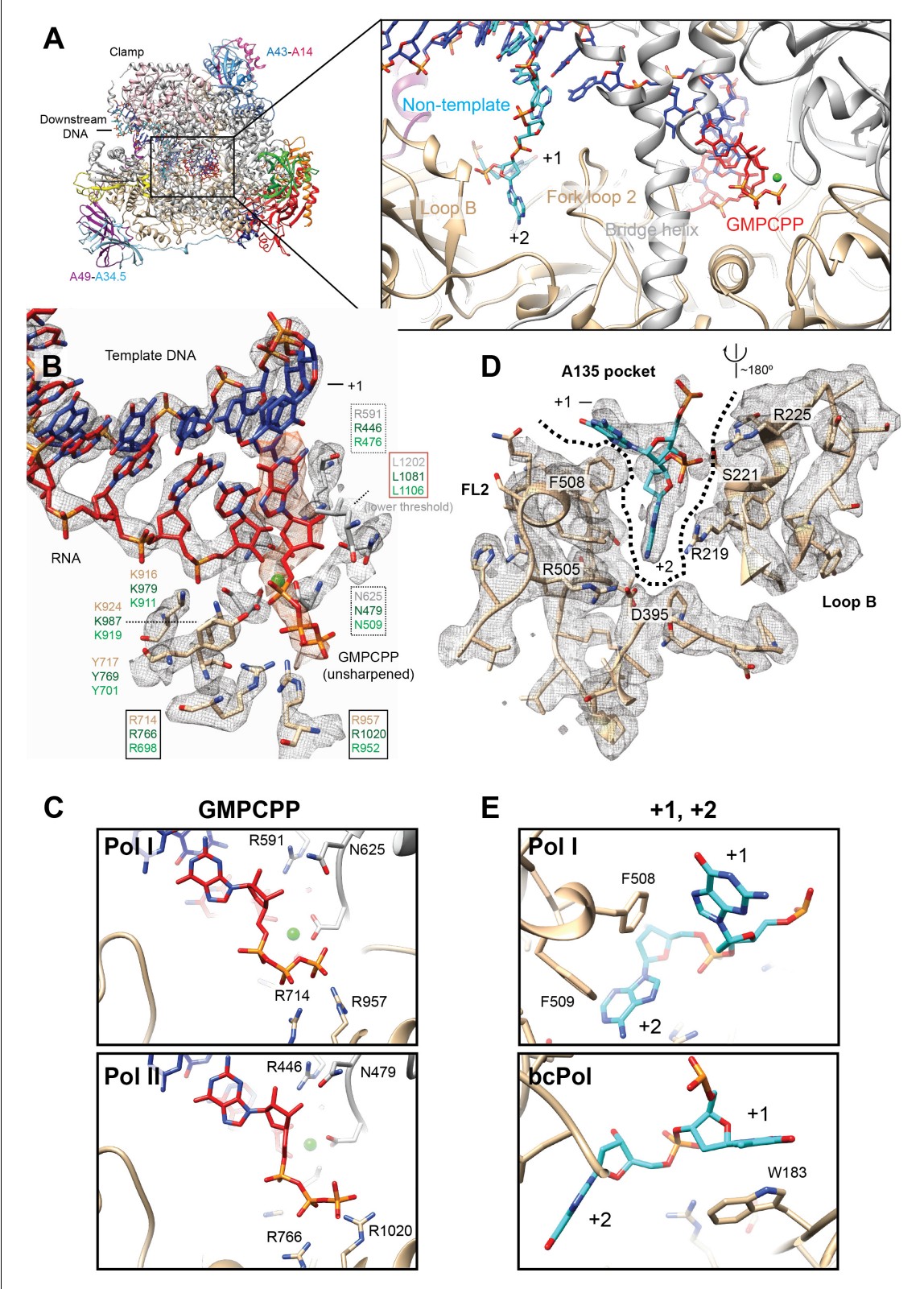

**Figure 5.** Interactions in the Pol I active site with GMPCPP, and the +1 and+2 bases from the non-template strand. (**A**) Pol I can bind the incoming nucleotide (GMPCPP) in the active site, while nucleotides of the opposite, non-template strand (+1 and+2), interact with the Fork loop two and Loop B. (**B**) GMPCPP is bound by conserved, identical residues in Pol I and Pol II. These include two arginines that interact with the phosphate (R714 and R957), a leucine from the trigger loop that stacks against the DNA base (L1202), and R591 and N625 which recognize the 2'- and 3'-OH groups, respectively.
*Figure 5 continued on next page*

**Figure 5 continued**

The 'gating tyrosine' (Y717), involved in RNA positioning during backtracking (*Cheung and Cramer, 2011*), and K916 and K924, which bind the 3'-end of the RNA are also indicated. Residues are shown in grey (A190) or tan (A135) for Pol I, while those in Pol II in dark green, and in Pol III in light green. Density for the DNA-RNA hybrid is from the sharpened, Pol I (core) EC (+GMPCPP) reconstruction, while the GMPCPP is from the same reconstruction but from the unsharpened/unmasked map. Density for L1202 is shown at a lower threshold. Residues are boxed according to their proposed role: black box, triphosphate binding; red box, nucleotide base stabilization; dashed box, NTP/dNTP discrimination. (C) Binding of GMPCPP is virtually identical in Pol I (top) and Pol II (bottom, PDB: 4a3j) (*Cheung et al., 2011*).(D) In the downstream edge of the transcription bubble, the +2 base of the NT strand is flipped into a pocket formed by Fork loop 2 (FL2) and loop B ('A135 pocket'). These elements interact with the nucleotide through R219, R225 and the conserved D395. (E) These interactions also position the +1 base next to F508 from FL2 (top), resembling the interaction of the +1 base with βW183 in bacterial Pol (bottom, PDB: 6alh) (*Kang et al., 2017*). See also *Figure 5—figure supplement 1*.

DOI: https://doi.org/10.7554/eLife.43204.010

The following figure supplement is available for figure 5:

**Figure supplement 1.** Conformational heterogeneity in the Pol I EC.
DOI: https://doi.org/10.7554/eLife.43204.011

As suggested by the conservation of residues in this region, the NTP is positioned in the 'A' site, as previously seen in Pol II (*Cheung et al., 2011*; *Wang et al., 2006*; *Westover et al., 2004*) and bacterial RNA polymerase (bcPol) (*Vassylyev et al., 2007*) (*Figure 5B*). Accordingly, the phosphate moiety is bound by two invariant arginine residues (A135 R714 and R957). In addition, the conserved A190 N625 and R591, which are involved in NTP/dNTP discrimination, come close to the 3'- and 2'-OH group, respectively. While the corresponding residue to N625 in Pol II (Rpb1 N479) has been shown to interact with either the 3'-OH (*Wang et al., 2006*) or the 2'-OH (*Cheung et al., 2011*), the invariant R591 (Rpb1 R446) interacts with the 2'-OH of the ribose in all structures. The NTP is maintained in the correct position by L1202 from the trigger loop, which interacts with the guanosine base. Only up to this residue, weak density can be observed, while the 'tip' loop (A190 residues 1203–1212) is unresolved. Overall, the positioning of the NTP substrate in the Pol I active site is virtually identical to that in Pol II (*Figure 5C*).

An interesting scenario is also observed opposite to the NTP binding site, where Pol I displays features similar to Pol II and bcPol. In both, the Pol I and Pol I* EC, the downstream edge of the transcription bubble is stabilized by interactions of Pol I with nucleotides + 1 and+2 from the NT strand. The +2 base is flipped into a pocket formed by elements from the A135 subunit, namely, the fork loop 2 (FL2) and loop B (*Tafur et al., 2016*) (*Figure 5D*). These two elements form a pocket ('A135 pocket') which resembles that formed by the β subunit ('β-pocket') in bcPol (*Zhang et al., 2012*). Whereas loop B exposes several positively charged residues towards the cavity of the A135 pocket that likely stabilize the phosphate backbone, a phenylalanine from the A135 FL2 (F508) appears to stack with the +1 base, in an analogous fashion as W183 from the bcPol β subunit (*Zhang et al., 2012*) (*Figure 5E*). Finally, the highly conserved D395 also interacts with the +2 base as in bcPol (β subunit D446) (*Vvedenskaya et al., 2014*) and Pol II (Rpb2 D399) (*Cheung and Cramer, 2011*), and probably also in Pol III (C128 D370). However, neither Pol II nor Pol III can form the equivalent interactions as in the A135 pocket because their corresponding loop B is differently positioned and far from the +1 and+2 bases. Interestingly, both sets of interactions are formed only when the DNA-binding cleft is completely closed and the jaw and clamp modules move towards each other (*Figure 5—figure supplement 1*). Thus, while formation of the EC involves the coarse movement of modules 1 and 2, nucleotide stabilization in the active site requires a more subtle, modular rearrangement.

## Discussion

Crystal and cryo-EM structures of Pol I in different functional states have revealed not only an overall conformational conservation compared to Pol II and Pol III, but have also shed light on the role of specific subunits, as well as the structural transitions from an inactive dimer to an actively transcribing enzyme (*Engel et al., 2018*). One of the main differences between the available Pol I structures is the position of A12.2C. In elongating Pol I, A12.2C is excluded from the active site, while no alternative position could be determined presumably because it is disordered. Here, we show that A12.2C can alternate between TFIIS-like and Rpb9-like positions depending on the presence of the

A49-A34.5 heterodimer. In the TFIIS-like position, A12.2C is positioned in the DNA-binding cleft and occludes the active site, which is incompatible with NTP incorporation but in accordance with RNA cleavage. When the cleft closes (and thereby clashes with A12.2C in the TFIIS-like position), A12.2C is excluded from the active site and can bind to the A135 ED1. Binding of A12.2C and heterodimer to the ED1 are mutually exclusive, as A12.2C and A34.5-Ct use overlapping binding sites. Exclusion of the heterodimer in this conformation is supported by the movement of the A135-Nt towards the HB domain, which blocks the interaction of the distal part of the A34.5-Ct with this domain. These results suggest a mechanism by which the surface of A135 (in particular, the ED1) plays a pivotal role in specific factor exchange in Pol I. Recent genetic studies have suggested that A12.2 may be involved in modulation of the movement of the jaw/lobe interface especially in the absence of A49, as the A49 linker and tWH domain appear to stabilize the closed conformation of Pol I when bound to DNA (*Darrière et al., 2018*). As the A12.2C binds to the A135 ED1, which sits next to the A135 lobe, the A12.2C might restrict the movement of the lobe. Thus, while A12.2N regulates the flexibility of the jaw, A12.2C could additionally regulate the movement of the lobe. Together, both A12.2 domains could therefore regulate cleft opening/closing of Pol I upon DNA binding, as well as binding to the +1 and+2 nucleotides in the non-template strand (see above). Restriction of movement of the A135 lobe by A12.2C might be important to maintain the closed state in the absence of A49, as in Pol I*. In contrast, when the heterodimer is present in the complex, A12.2 might destabilize the EC as it can only occupy the TFIIS-like site, thereby preventing cleft closure (*Appling et al., 2018*). In this scenario, A49 could play an important role in maintaining a narrow cleft, which would also explain (in addition to the direct interaction of their N-terminal domains with A12.2) the stimulatory role of the heterodimer on A12.2-mediated RNA cleavage (*Geiger et al., 2010*).

In vivo, heterodimer association to Pol I might offer an additional layer of regulation of rDNA transcription (*Figure 6*). The proportion of initiation-competent Pol I molecules in the cell has been proposed to represent those Pol I particles bound to initiation factor Rrn3 (*Milkereit and Tschochner, 1998*). In contrast, the number of Pol I* particles in the cell could represent a population of actively transcribing DNA-bound Pol I, but also a pool of pre-active Pol I that can readily initiate transcription upon heterodimer binding and Rrn3 recruitment (in contrast to Pol I dimers, which appear to be a storage form of the enzyme (*Torreira et al., 2017*)). The number of initiation-competent Pol I molecules could be thus regulated not only by Pol I homo-dimerization and association with Rrn3, but also by changes in the heterodimer concentration in the nucleolus, thereby controlling the ratio of Pol I to Pol I*. Nutrient-dependent regulation of nucleolar localization of the mammalian A49-A34.5 homolog PAF53-PAF49 has been observed (*Penrod et al., 2015*). PAF49 (A34.5 counterpart) accumulates in the nucleolus in growing cells but disperses to the nucleoplasm upon serum starvation (*Yamamoto et al., 2004*). In yeast, A34.5 is maintained in the nucleolus by its association with A49 (but also contains a nucleolar localization signal in its C-terminal region), and A49 is required for the high loading rate of Pol I onto rDNA (*Albert et al., 2011*). Human PAF53-PAF49 can substitute the A49-A34.5 heterodimer in vivo (*Albert et al., 2011*) suggesting a conserved function (and possibly regulation). Regulation of heterodimer binding to Pol I might also explain why promoter association of Pol I-Rrn3 complexes is low upon nutrient starvation even when the concentration of such complexes is relatively high (*Torreira et al., 2017*); the levels of the heterodimer might further regulate Pol I initiation rates.

In addition, the release of the heterodimer from the enzyme would also allow the binding of elongation factors to Pol I. Pol I has been shown to bind to elongation factor Spt5 directly (*Viktorovskaya et al., 2011*) and its activity is affected by Spt4/5 in vivo (*Anderson et al., 2011*). In the Pol I EC, canonical binding of Spt4/5 (as in the Pol II EC) is precluded by the A49 tWH (*Tafur et al., 2016*), as it occupies a position equivalent to the KOW1-L1 domain of Spt5, and by the A49 linker helix spanning the cleft, which clashes with the N-terminal region of Spt5 (*Figure 6—figure supplement 1*) (*Bernecky et al., 2017*; *Ehara et al., 2017*). Interestingly, Spt5 interacts physically and genetically with A49, suggesting a functional interplay between these proteins (*Viktorovskaya et al., 2011*). Paf1C, another elongation factor, has also been shown to stimulate Pol I transcription in vivo and in vitro (*Zhang et al., 2009*; *Zhang et al., 2010*). Paf1C binds to Pol II on the outer surface of subunit Rpb2 (Pol II counterpart of A135) including the Rpb2 ED2 and lobe (*Vos et al., 2018*; *Xu et al., 2017*). In this position, it clashes and competes with TFIIF for Pol II binding (*Xu et al., 2017*). Heterodimer dissociation from Pol I could potentially free the binding site for both Spt4/5 and Paf1C in a mechanism that could be akin to the transition from initiation to

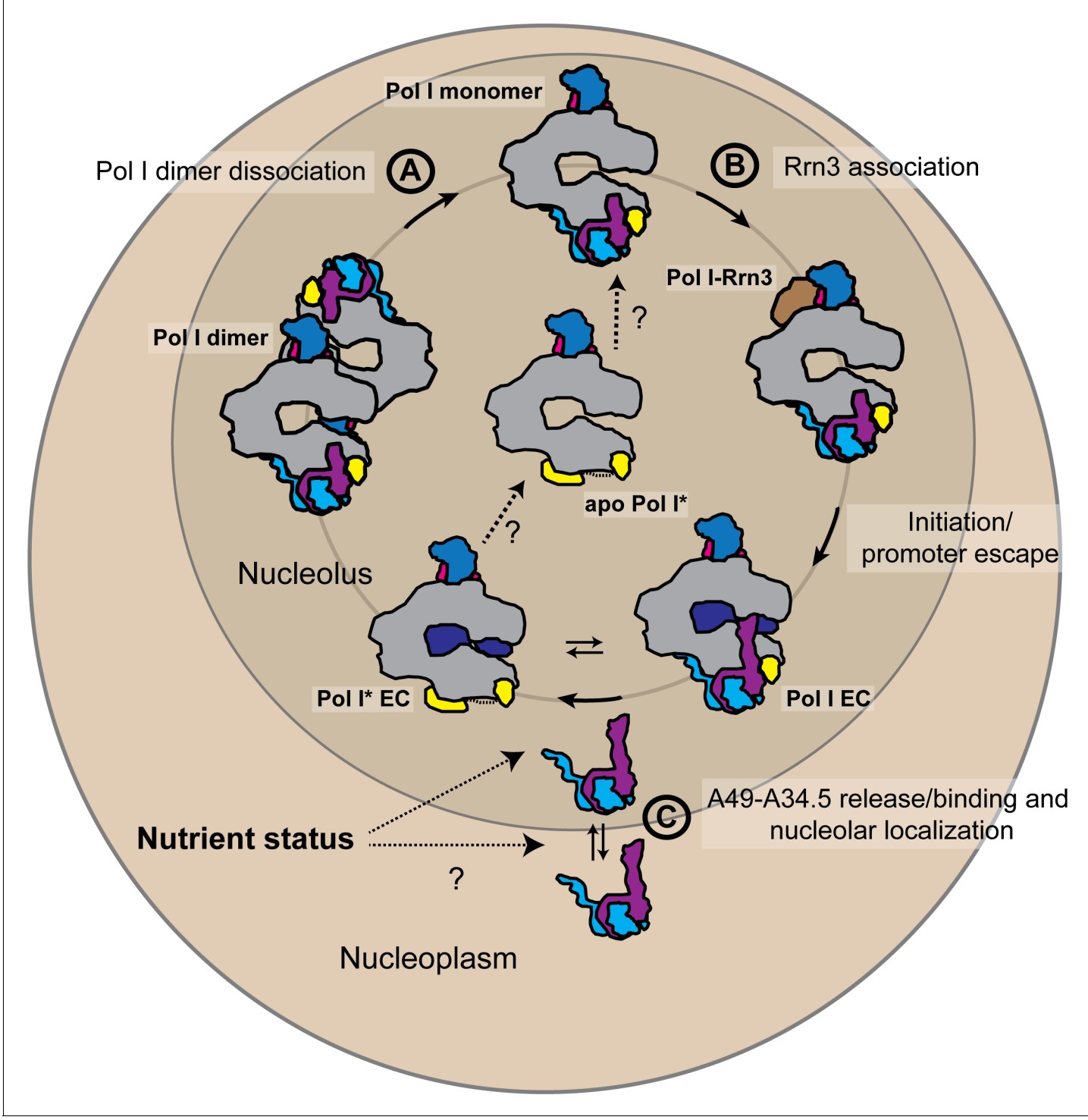

**Figure 6.** Schematic representation of the possible physiological role of the A49-A34.5 heterodimer in the regulation Pol I activity. The pool of initiation-competent Pol I particles is controlled by Pol I homo-dimerization (**A**) and binding of Rrn3 to monomeric Pol I (**B**). After transcription initiation and promoter escape, during elongation, Pol I can alternate between Pol I and Pol I* conformations. Release of the A49-A34.5 heterodimer would allow the recruitment of elongation factors (**C**). After dissociating from DNA, Pol I* could bind to the A49-A34.5 heterodimer to replenish the pool of initiation-competent Pol I monomers. The concentration of A49-A34.5 heterodimer in the nucleolus might be also regulated by the nutrient status of the cell as in the mammalian system. Regulated localization of the A49-A34.5 heterodimer would serve to alter the ratio of Pol I to Pol I* in the nucleolus, thereby controlling the initiation rate on the rDNA. See also *Figure 6—figure supplement 1*.

DOI: https://doi.org/10.7554/eLife.43204.012

*Figure 6 continued on next page*

*Figure 6 continued*

The following figure supplement is available for figure 6:

**Figure supplement 1.** A49-A34.5 heterodimer release frees the binding site for Spt4/5 and Paf1C.

DOI: https://doi.org/10.7554/eLife.43204.013

elongation in Pol II: while TFIIE (A49 tWH) blocks the Spt4/5 binding site, TFIIF (A49-A34.5 dimerization domain) occupies the binding site of part of Paf1C (*Vos et al., 2018*; *Xu et al., 2017*) (*Figure 6—figure supplement 1*). Thus, binding of elongation factors is mutually exclusive with the presence of initiation factors. Therefore, in Pol I, factor exchange during the transition from initiation to elongation could be accommodated more readily just by the release of the heterodimer and switching to the Pol I* form. In this scenario, A12.2 might further prevent re-association of the heterodimer. A similar allosteric transition during promoter escape mediated by the heterodimer, Spt5 and the stalk has been previously proposed for Pol I (*Beckouët et al., 2011*). Because we could not observe any effect of free A12.2C on heterodimer binding to Pol I in vitro, release of the heterodimer in vivo might be directly induced by Spt4/5 and Paf1C.

## Materials and methods

### Pol I EC-GMPCPP complex formation

Endogenous Pol I was purified from yeast cells as previously described (*Moreno-Morcillo et al., 2014*). Pol I was incubated with a 38 base pair transcription scaffold containing an 11 nucleotide mismatch bubble and a 20 nucleotide RNA as used previously for formation of the Pol I EC (*Tafur et al., 2016*). The complex was incubated for 1 hr at 4°C in 15 mM HEPES-NaOH (pH 7.5), 150 mM ammonium sulfate, 1 mM $MgCl_2$, 1 mM GMPCPP (Jena Bioscience) and 10 mM DTT. The sample was diluted to ~0.1 mg/mL in the same buffer immediately before grid freezing.

### Cryo-EM sample preparation

2.5 µL of sample was deposited on a freshly glow-discharged cryo grid (R 2/1 + 2 nm carbon, Quantifoil), incubated for 30 s, and blotted for 3 s (with a blotting force of '3'), at 100% humidity and 4°C in a Vitrobot Mark IV (FEI). Grids were stored in liquid nitrogen until data collection.

### Cryo-EM data collection

5768 micrograph movies were collected on a FEI Titan Krios at 300 keV through a Gatan Quantum 967 LS energy filter using a 20 eV slit width in zero-loss mode. The movies were recorded on a Gatan K2 direct electron detector, at a nominal magnification of 135,000x corresponding to a pixel size of 1.04 Å in super resolution mode, using Serial EM. Movies were collected in 40 frames with defocus values from −0.75 to −2.5 µM, with a dose of 0.9775 $e^-$ $Å^{-2}$ $s^{-1}$ per frame for 16 s.

### Cryo-EM data processing

Movies were aligned, motion-corrected and dose-fractionated using MotionCor2 (*Zheng et al., 2017*). Contrast transfer function (CTF) estimation was done using CTFFIND4 (*Rohou and Grigorieff, 2015*). All processing steps were performed in Relion 2.0 (*Kimanius et al., 2016*) unless otherwise indicated. Resolution estimates reported are those obtained after masking and B-factor sharpening (Relion post-processing). Data were divided in five batches to increase processing speed. For each batch, autopicking was followed by a 2D classification step (with data downsized five times) to remove contamination and damaged particles. Good classes were selected, re-extracted and unbinned, and refined against the Pol I EC (PDB: 5m5x) low pass filtered to 40 Å. Then, a 3D classification step was performed without alignment. For all batches the same procedure was followed, except for batch 5, in which 3D classification was performed with data downsized five times. Classes were selected based on the width of the cleft, the position of the clamp, and the DNA-RNA scaffold density, and grouped by similarity. Refinement of the pooled particles with closed cleft and strong DNA-RNA density revealed an extra density and streaky, weak density for the A49-A34.5 heterodimer. To resolve this region, a masked classification was performed. This yielded a class with high resolution in the extra density, allowing the unambiguous assignment of the A12.2 C-terminal

domain (A12.2C). Based on these results, all other pooled classes were classified with a mask on this area. Particles were merged depending on whether they showed density for the A49-A34.5 hetero-dimer (Pol I) or the A12.2C without A49-A34.5 (Pol I*). During the process, additional bad particles were discarded by global 3D classification without a mask nor alignment. After refinement of all good particles for Pol I and Pol I*, additional classification steps were performed to increase the resolvability of the active site. For Pol I* particles, a 3D classification step with a mask on the core and DNA-RNA hybrid yielded a class (182,488 particles) with a better density for GMPCPP, which could be refined to 3.18 Å resolution. An apo form of Pol I* consisting of 73,660 particles was obtained during a global classification step of the initial subset with a closed cleft and strong DNA-RNA density, and was refined to 3.21 Å resolution. For the pooled Pol I particles, a global 3D classification step yielded a class with a closed clamp (EC) and a class bound to DNA-RNA with a slightly more open clamp. The latter was classified one more round, which gave a class in an EC conformation. These particles were merged with the EC particles from the previous 3D classification step, refined (consensus Pol I EC) and classified with a mask on the core, the full DNA-RNA scaffold and the linker helix of A49, which yielded a class with strong GMPCPP density (30,232 particles) that was refined to 3.42 Å resolution. As both Pol I EC and Pol I* EC reconstructions were very similar in the active site, EC particles were merged and classified using different masks. Masked classification based on the full DNA scaffold and rudder produced one class (34,475 particles) with improved density for the upstream DNA duplex and revealing the path of the single stranded non-template strand (ssNT), which was refined to 4.0 Å resolution (without post-processing). Classification based on the core and DNA-RNA scaffold revealed different states differing in the width of the cleft, base flipping at position +2, presence of the GMPCPP and conformation of the trigger loop (shown in *Figure 5— figure supplement 1*). One of these classes (Pol I (core) EC +GMPCPP), which showed better density for GMPCPP, the +2 base and A190 L1202 was refined to 3.18 Å resolution (54,017 particles). Local resolution was estimated with Blocres (*Cardone et al., 2013*).

## Model building and refinement

Previous Pol I structures in its apo (PDB: 4c3i and 4c2m) and elongating (PDB: 5m5x) forms were used as starting models. The initial placement of GMPCPP in the active site was based on its position in a Pol II EC with bound GMPCPP (*Wang et al., 2006*) (PDB: 2e2j and 4a3j). Initially, the model for the Pol I (core) EC (+GMPCPP) was built in COOT (*Emsley and Cowtan, 2004*) and real-space refined in PHENIX (*Adams et al., 2010*). This model was then rigid body fitted in the Pol I* or Pol I EC (+GMPCPP) maps in UCSF Chimera, further adjusted in COOT, and real-space refined again in PHENIX. For Pol I*, residues 66–125 from A12.2 were taken from the apo crystal structure (PDB: 4c3i), fitted to the density and manually adjusted. The A12.2 linker region was deleted afterwards. Agreement between maps and models was estimated in PHENIX. Model quality was assessed with Molprobity (*Chen et al., 2010*).

## Expression, purification and labeling of recombinant A49-34.5

The cDNA of *S. cerevisiae* of rpa49 and rpa34 was codon-optimized for bacterial expression hosts and synthesized by GenScript. The two genes were cloned into separate ORFs in a pRSF Duet expression vector (Novagen) for co-expression. Codons for native cysteine residues were exchanged for alanine by mutagenesis PCR. Another mutation in A49 was introduced resulting in A140C to introduce a fluorescent label at this position. The construct was expressed in *E. coli* BL21 (DE3) Star in TB media by incubation with shaking at 37°C until an $OD_{600nm}$ of 0.8 was reached. The temperature was shifted to 18°C and expression was induced by addition of 0.05 mM IPTG at an $OD_{600nm}$ of 1 to 1.2. After 16 hr, cells were harvested by centrifugation. Cells were lysed using an enzymatic-chemical approach by resuspending in a buffer containing lysozyme, DNaseI and Triton-X 0.1% in 50 mM Tris (pH 7.5), 300 mM NaCl, 10 mM $MgCl_2$, 10 mM β-mercapto ethanol, and 5 mM imidazole. The mixture was stirred at 4°C for 2–4 hr. The lysate was cleared by centrifugation (45,000 g for 90 min at 4°C) and the supernatant incubated with 5 to 10 mL Ni-NTA beads (QIAGEN) while rotating for 1 hr at 4°C. The beads were collected by gravity flow in a Biorad column and washed with 100 mL of washing buffer (50 mM Tris (pH 7.5), 500 mM NaCl, 10 mM β-mercapto ethanol, and 10 mM imidazole). Bound protein was eluted with 10–20 mL elution buffer (50 mM Tris (pH 7.5), 300 mM NaCl, 10 mM β-mercapto ethanol, and 300 mM imidazole). The elution fraction was dialyzed

overnight against SP Buffer A (50 mM Tris (pH 7.5), 100 mM NaCl, 10 mM DTT). The next day, the protein solution was loaded onto a 5 mL HiTrap SP column (GE Healthcare) and eluted into 1 mL fractions with a 10 CV gradient from 100 to 1000 mM NaCl in 50 mM Tris (pH 7.5) with 10 mM DTT. Elution fractions of the major peak were analyzed by SDS-PAGE, combined, concentrated, and loaded onto a Superdex 200 (120 mL, GE Healthcare) equilibrated in 25 mM HEPES (pH 7.4), 150 mM NaCl, and 0.5 mM TCEP. The peak fractions were analyzed by SDS-PAGE, combined and concentrated. Protein identity was confirmed by mass spectrometry. The purified heterodimer was directly labeled with maleimide-functionalized Alexa Fluor 594 that was freshly dissolved at 10 mM in DMSO. The dye was added slowly to the protein solution with a final ratio of in 1:10 (protein:dye). The mixture was incubated overnight in the dark while shaking (800 rpm) at 4°C. The reaction was quenched by addition of 10 mM DTT and unreacted dye molecules were removed by size-exclusion chromatography (Superdex 200, 24 mL, GE Healthcare) equilibrated in reconstitution buffer (50 mM ammonium sulfate, 25 mM HEPES (pH 7.4), 10 mM $MgCl_2$, 10 mM DTT). Labeling efficiency was determined by UV-VIS measurements of protein (280 nm) and dye absorbance.

## Expression and purification of A12.2C

The part of cDNA of *S. cerevisiae* rpa12 coding for the A12.2 C-terminal domain (residues 79 to 125) was cloned into a pET24a expression vector with an N-terminal 6xHis tag followed by a TEV cleavage site. The construct was expressed in *E. coli* BL21 (DE3) Star in TB media by shaking at 37°C until an $OD_{600nm}$ of 0.8 was reached. Expression was induced by adding 0.5 mM IPTG and continued at 37°C for 4 hr. Cells were harvested, re-suspended in lysis buffer (50 mM Tris (pH 8), 500 mM NaCl, 10 mM β-mercaptoethanol, 5 mM imidazole, pH 8), and lysed by sonication. The cleared lysate was incubated with Ni-NTA beads (QIAGEN) for 1 hr at 4°C. Beads were washed with 50 mM Tris (pH 8), 500 M NaCl, 10 mM β-mercapto ethanol, and 10 mM imidazole and incubated in 30 mL wash buffer with 1.5 mg of TEV overnight. The cleaved protein was concentrated to about 2 mL and loaded onto a Superdex 75 (GE Healthcare) equilibrated in 50 mM HEPES (pH 7.5), 150 mM NaCl, and 0.5 mM TCEP. The major peak was collected, analyzed by SDS-PAGE, combined and concentrated. Protein identity was confirmed by mass spectrometry.

## Expression and purification of Pol I*

The yeast strain Y2670 harboring Pol I Δrpa49 was generously provided by Herbert Tschochner (Universität Regensburg) (*Pilsl et al., 2016*). The mutant strain was expressed and purified analogous to the wild type Pol I yielding pure Pol I*.

## Fluorescence polarization measurements

Purified Pol I* was incubated with labeled heterodimer at different concentrations overnight at 4°C in 150 mM ammonium sulfate, 15 mM HEPES (pH 7.5) and 10 mM DTT. For measurements using the Pol I* EC, Pol I* was incubated with an equimolar concentration of the same transcription scaffold used for the cryo-EM data for 1 hr at 4°C, previous to the overnight incubation. For the experiments using the A12.2C, recombinant A12.2C was incubated with the labeled Pol I EC for 30 min at room temperature.

Fluorescence polarization of A49(A140C)−34.5 heterodimer labeled with Alexa Fluor 594 was measured on a Jasco FP-6000 fluorometer equipped with polarization filters in a 150 μL volume with a final concentration of 100 nM of the labeled species. Fluorescence intensities at different polarization angles were measured at 594 nm excitation (2.5 nm bandwidth) and 625 nm emission (10 nm bandwidth) wavelengths. The anisotropy was calculated for the free and bound heterodimer by using an excess of Pol I* bound to DNA.

## Accession numbers

Models have been deposited in the PDB with codes: 6HKO (Pol I EC +GMPCPP), 6HLQ (Pol I* EC +GMPCPP), 6HLR (Pol I (core) EC +GMPCPP), and 6HLS (apo Pol I*). Cryo-EM maps have been deposited in the EMDB with codes: EMD-0238 (Pol I EC +GMPCPP), EMD-0239 (Pol I* EC +GMPCPP), EMD-0240 (Pol I (core) EC +GMPCPP), EMD-0241 (apo Pol I*) and EMD-0242 (Pol I EC +GMPCPP (upstream DNA focused)).

## Acknowledgements

YS, LT, RW and CWM acknowledge support by the ERC Advanced Grant (ERC-2013-AdG340964-POL1PIC). LT acknowledges support by the EMBL International PhD program. JH acknowledges EMBO for a postdoctoral long-term fellowship (ALTF 372–2017). We thank Matthias Vorländer, Florence Baudin, Herman KH Fung, Kathryn Perez and Vladimir Rybin for advice and discussion.

## Additional information

### Funding

| Funder | Grant reference number | Author |
|---|---|---|
| European Commission | ERC-2013-AdG340964-POL1PIC | Lucas Tafur<br>Yashar Sadian<br>Jonas Hanske<br>Rene Wetzel<br>Christoph W Müller |
| European Molecular Biology Organization | ALTF 372-2017 | Jonas Hanske |

The funders had no role in study design, data collection and interpretation, or the decision to submit the work for publication.

### Author contributions

Lucas Tafur, Data curation, Formal analysis, Validation, Investigation, Visualization, Methodology, Writing—original draft; Yashar Sadian, Data curation, Formal analysis, Investigation, Visualization; Jonas Hanske, Data curation, Investigation, Writing—review and editing; Rene Wetzel, Responsible for wild type and mutant yeast fermentation and wild type and mutant RNA polymerase I purification; Felix Weis, Data curation, Investigation; Christoph W Müller, Supervision, Funding acquisition, Project administration, Writing—review and editing

### Author ORCIDs

Christoph W Müller (iD) https://orcid.org/0000-0003-2176-8337

### Decision letter and Author response

Decision letter https://doi.org/10.7554/eLife.43204.032
Author response https://doi.org/10.7554/eLife.43204.033

## Additional files

### Supplementary files

• Transparent reporting form
DOI: https://doi.org/10.7554/eLife.43204.014

### Data availability

Coordinates and cryo-EM maps have been deposited with the PDB and EMDB, respectively.

The following datasets were generated:

| Author(s) | Year | Dataset title | Dataset URL | Database and Identifier |
|---|---|---|---|---|
| Tafur L, Sadian Y, Weis F, Muller CW | 2018 | Pol I (core) EC + GMPCPP | https://www.ebi.ac.uk/pdbe/entry/emdb/EMD-0240 | Electron Microscopy Data Bank, EMD-0240 |
| Tafur L, Sadian Y, Weis F, Muller CW | 2018 | Pol I (core) EC + GMPCPP | https://www.rcsb.org/structure/6HLR | Protein Data Bank, 6HLR |
| Tafur L, Sadian Y, Weis F, Muller CW | 2018 | Pol I EC + GMPCPP | https://www.ebi.ac.uk/pdbe/entry/emdb/EMD-0238 | Electron Microscopy Data Bank, EMD-0238 |

| Tafur L, Sadian Y, Weis F, Muller CW | 2018 | Pol I EC + GMPCPP | https://www.rcsb.org/structure/6HKO | Protein Data Bank, 6HKO |
|---|---|---|---|---|
| Tafur L, Sadian Y, Weis F, Muller CW | 2018 | Pol I* EC + GMPCPP | http://www.ebi.ac.uk/pdbe/emdb/EMD-0239 | Electron Microscopy Data Bank, EMD-0239 |
| Tafur L, Sadian Y, Weis F, Muller CW | 2018 | Pol I* EC + GMPCPP | https://www.rcsb.org/structure/6HLQ | Protein Data Bank, 6HLQ |
| Tafur L, Sadian Y, Weis F, Muller CW | 2018 | Apo Pol I* | https://www.ebi.ac.uk/pdbe/entry/emdb/EMD-0241 | Electron Microscopy Data Bank, EMD-0241 |
| Tafur L, Sadian Y, Weis F, Muller CW | 2018 | Apo Pol I* | https://www.rcsb.org/structure/6HLS | Protein Data Bank, 6HLS |

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
