## [Decision Letter]

[Editors’ note: a previous version of this study was rejected after peer review, but the authors submitted for reconsideration. The first decision letter after peer review is shown below.]

Thank you for submitting your work entitled "Structural rearrangement of TFIIS- and TFIIE/TFIIF-like subunits in RNA polymerase I transcription complexes" for consideration by *eLife*. Your article has been reviewed by three peer reviewers and the evaluation has been overseen by a Reviewing Editor and a Senior Editor. The following individual involved in review of your submission has agreed to reveal their identity: Alessandro Vannini (Reviewer #3).

Our decision has been reached after consultation between the reviewers. Based on these discussions and the individual reviews below, we regret to inform you that your work in its present form will not be considered further for publication in *eLife*. However, as described below, we would be open to considering a new manuscript that takes into account the concerns and suggestions articulated below.

This paper describes cryo-EM studies of RNA polymerase I (Pol I) that provide new information on several aspects of the A49-A34.5 heterodimer, which is the functional equivalent of the Pol II TFs, TFIIE and TFFIIF, and A12.2, the functional equivalent of TFIIS. The new structures in this manuscript were obtained from a preparation of Pol I bound to an NTP analogue, GMPCPP, in the absence of MgCl_2_. By sorting and classifying subsets of particles, the authors were able to identify distinct species representing the Pol I elongation complex (EC) + GMPCPP, Pol I core EC + GMCPP, Apo Pol I* (i.e. lacking the A49-A34.5 heterodimer) and Pol I* EC + GMCPP. By comparing these structures with previous structure of all three eukaryotic RNA polymerases, the authors draw several new insights. One interesting observation is the relocation of the A49-A34.5 heterodimer, which is exchanged for the A12 subunit and thus promotes shift between a Pol I and Pol II-like conformation. The structures also provide additional details on the interaction of Pol I with dNTP and stabilization of the transcription bubble.

All three reviewers appreciated the technical quality of the structural studies and the implications of some of the findings, in particular the role of heterodimer exchange with A12. However, it was agreed that the manuscript in its current form is written in a way that is accessible to experts only and presumes a high degree of familiarity with the Pol I literature in particular, as well as with Pol II and III. A significant number of new structural observations are described, although they are not all well-connected in terms of the questions they are addressing. The dimer exchange model was considered one interesting aspect of this new set of structures; however, this model needs to account more explicitly for genetic data suggesting that the heterodimer is present throughout the gene. Additional experimental data supporting this model could also strengthen the paper. A significant revision and refocusing of the paper would be needed to make the work described here more appropriate for the general readership of *eLife*.

*Reviewer #1:*

This paper describes cryo-EM studies of RNA polymerase I (Pol I) that provide new information on several aspects of the A49-A34.5 heterodimer, which is the functional equivalent of the Pol II TFs, TFIIE and TFFIIF, and A12.2, the functional equivalent of TFIIS. There have been multiple structures determined of Pol I by cryo-EM and x-ray crystallography, including several by the senior author. The new structures in this manuscript were obtained from a preparation of Pol I bound to an NTP analogue, GMPCPP, in the absence of MgCl_2_. By sorting and classifying subsets of particles, the authors were able to identify distinct species representing the Pol I elongation complex (EC) + GMPCPP, Pol I core EC + GMCPP, Apo Pol I* (i.e. lacking the A49-A34.5 heterodimer) and Pol I* EC + GMCPP. By comparing these structures with previous structure of all three eukaryotic RNA polymerases, the authors draw several new insights. Chief among these is that binding of the A49-A34.5 heterodimer and A12.2C are incompatible, and thereby gate Pol I conformation shifts between a Pol I and Pol II-like conformation. There are also additional insights into the detailed interactions that stabilize the transcription bubble.

Overall, the manuscript reads like a collection of very detailed structural information on different Pol I conformations with no clear direction. The questions to be addressed are not framed clearly, so the reader is left largely with an unfocused catalogue of structural details without a clear sense of why they matter. The paper is also written in a way that will be accessible primarily to those who work on Pol I and are already familiar with the structure and prior publications. The complicated and unfortunate naming system of Pol I subunits (no fault of the authors) makes the manuscript an even tougher slog, especially without sufficient graphic overviews to guide the reader. In the end, while the authors speculate about the meaning of various observations, for example about the role of the heterodimer in regulating binding of elongation factors, these ideas are not tested experimentally. While these new structures will be of interest to specialists, there are insufficient insights that will be of interest to the general readership of *eLife*.

Specific points:

The interpretation of additional density (Figure 1—figure supplement 3) that "connects to the DML toward the A12.2 linker,” etc.) is not at all convincing or even clear as described. It is difficult to tell where in the structure the density is located due to poor labeling. Moreover, the legend states that this density is only visible at low contour levels. Without further clarification of which residues this might correspond to, this should be omitted.

*Reviewer #2:*

In this manuscript, the authors bring forward some detailed, reasonably high resolution models for positions in and around the active site of RNA polymerase I. They present a catalogue of comparisons to the other eukaryotic polymerases as well as the prokaryotic enzyme. The big picture is really quite compelling: These enzymes work via the same mechanism, but they have apparently evolved unique properties that are reflected in the structures. Big picture: I like the manuscript.

There are some significant challenges, though, that can be addressed and would make the story seem better supported.

1) The authors have a robust understanding of the three polymerases, but their comparison of similarities could use more clarification for the non-expert reader.

2) This is a feature of structural biology, but this manuscript seems to make substantial mechanistic conclusions from the structural models. This was particularly evident in the section describing the motions in the +1 and +2 positions. Throughout, there should be more care taken to indicate that these are hypotheses.

3) Does the heterodimer really leave the enzyme during transcription? The manuscript draws on some literature, but is this a well-supported view? Can there be tests?

4) A commentary on dynamics might be interesting. The discussion of these subunits being exclusive, then swapping with transcription factors (Discussion section) would benefit from a consideration of dynamics in solution (or in living cells).

*Reviewer #3:*

The manuscript by Tafur et al. reports cryo-EM structures of elongating RNA Polymerase I in presence of a nucleotide analog, revealing that the heterodimer A49-A34.5 has been expelled from the complex and the C-term domain of A12.2 is relocated on a surface that was previously occupied by the heterodimer.

The cryo-EM reconstructions are impeccable and of high quality. The main conclusions are justified by the structural findings. This is potentially very interesting also because a form of the enzyme lacking A49-34.5 has been identified in vivo.

I strongly support publication, upon minor textual revisions and clarification of few minor issues

1) Can the author monitor "ejection" of the heterodimer upon incubation with GMPCC? This would be nice to see. This could probably be done on immobilised DNA beads and the composition of Pol I monitored over time in presence of GMPCC.

2) Results section “Cryo-EM structures of the GMPCPP-bound Pol I elongation complex (EC): this is then the authors elaborate on how this apo PolI* is formed? And why wasn't this observed in previous reconstruction of elongating Polymerase I? Could it be that Pol I* is a less stable elongation complex and partially loses the DNA scaffold after ejection of the heterodimer? I think that an in vitro assay (and possibly re-titrating in recombinant heterodimer) would add a lot to the manuscript and help understand the mechanism.

3) The first paragraph of the Introduction is difficult to follow maybe rewrite more concisely?

4) In the Accession numbers section, are the complexes named properly? Which one is the apo PolI*?

5) Last sentence of the Discussion: the fact the general catalytic mechanism in RNA Polymerase is conserved was already clear. I would suggest that the molecular determinants/regions involved in catalysis performs similar functions and this is now successfully proven.

[Editors’ note: what now follows is the decision letter after the authors submitted for further consideration.]

Thank you for submitting your article "The cryo-EM structure of a 12-subunit variant of RNA polymerase I reveals dissociation of the A49-A34.5 heterodimer" for consideration by *eLife*. Your article has been reviewed by three peer reviewers, one of whom is a member of our Board of Reviewing Editors, and the evaluation has been overseen by John Kuriyan as the Senior Editor. The following individual involved in review of your submission has agreed to reveal his identity: Alessandro Vannini (Reviewer #2).

The reviewers have discussed the reviews with one another and the Reviewing Editor has drafted this decision to help you prepare a revised submission.

Summary:

This is a revision of a previous submission describing cryo-EM studies of RNA polymerase I (Pol I). The revised manuscript is improved in that it focuses on exchange of the A49-A34.5 heterodimer and A12.2, and the differences between Pol I and Pol I*. By sorting and classifying subsets of particles, the authors were able to identify distinct species of Pol I core and Pol I*. Chief among the conclusions is that binding of the A49-A34.5 heterodimer and A12.2C are incompatible, and thereby gate Pol I conformation shifts between a Pol I and Pol II-like conformation. The addition of binding data on A49-A34.5 heterodimer and A12.2C is, in principle, a positive addition and was requested in the previous review. However, there were issues with the binding data that must first be addressed before the manuscript is ready for publication.

Essential revisions:

1) The binding curves and analysis are problematic. According to the methods section, labeled heterodimer was present at 100 nM concentration in the binding studies. If this was indeed the concentration, the equation for the binding isotherm is not valid, as this approximation assumes that the labeled species is present at much lower concentration than Pol I. Either the binding data should be re-analyzed with a quadratic binding equation or the experiments should be repeated at much lower labelled heterodimer concentration.

2) The proposal that sequential binding of the N- and C-terminal domains of the heterodimer cannot explain apparent cooperativity, as the experiment only monitors binding of single monomers. It is possible that the analysis of the data as suggested above, a more complex model or a repeat of the experiment under different conditions could sort this out.

3) There was not enough information given to evaluate the competition/exchange experiment. If the heterodimer indeed binds with a 14 nM K_D_ and the complex with Pol I was present at 100 nM concentration, almost all of the proteins would be in a complex. Exchange with A12.2C would depend upon the off rate of A49-A34.5 and the on rate for A 12.2C. The complex was incubated with A12.2C for 30 minutes. Were longer time points measured? What is the K_D_ of A12.2C for Pol I? Might there be different conditions under which dimer exchange would be observed? The ultimate conclusion, that the heterodimer must first dissociate in order for A12.2C to bind is surely correct in light of the structure, but it is important that the paper has binding data that show this. A good control for the experiment would be exchange with unlabeled heterodimer, as this would provide a good benchmark. Similarly, the authors should consider repeating the GMCPP experiment at lower concentration of Pol I* – heterodimer complex, as it is possible the complex would dissociate if the concentrations were not so far above the (apparent) K_D_.

---

## [Author Response]

[Editors’ note: the author responses to the first round of peer review follow.]

[…] All three reviewers appreciated the technical quality of the structural studies and the implications of some of the findings, in particular the role of heterodimer exchange with A12. However, it was agreed that the manuscript in its current form is written in a way that is accessible to experts only and presumes a high degree of familiarity with the Pol I literature in particular, as well as with Pol II and III. A significant number of new structural observations are described, although they are not all well-connected in terms of the questions they are addressing. The dimer exchange model was considered one interesting aspect of this new set of structures; however, this model needs to account more explicitly for genetic data suggesting that the heterodimer is present throughout the gene. Additional experimental data supporting this model could also strengthen the paper. A significant revision and refocusing of the paper would be needed to make the work described here more appropriate for the general readership of eLife.

We would like to thank all reviewers for their comments and suggestions. We believe that their input has significantly improved our original manuscript. The revised manuscript now focuses on the exchange of the A49-A34.5 heterodimer with the C-terminal domain of subunit A12.2 (A12.2C) as observed in the Pol I* structure, and we discuss our structural findings in the context of known biochemical and genetic data. In addition, we have developed an assay to track the binding and dissociation of the heterodimer to the Pol I core by fluorescence polarization. We observe that under our experimental conditions, the heterodimer binds with high affinity and cannot be displaced by either free A12.2C or GMPCPP. Thus, binding of A12.2C can only occur once the A49-A34.5 heterodimer has been released from the enzyme. Moreover, the features of the active site (GMPCPP binding, and interactions with the non-template strand) are now discussed in the context of the Pol I* variant. Finally, we have revised our figures and regrouped them to increase the clarity of the paper. A detailed answer to each of the reviewer’s comments is listed below.

Reviewer #1:[…] Overall, the manuscript reads like a collection of very detailed structural information on different Pol I conformations with no clear direction. The questions to be addressed are not framed clearly, so the reader is left largely with an unfocused catalogue of structural details without a clear sense of why they matter. The paper is also written in a way that will be accessible primarily to those who work on Pol I and are already familiar with the structure and prior publications. The complicated and unfortunate naming system of Pol I subunits (no fault of the authors) makes the manuscript an even tougher slog, especially without sufficient graphic overviews to guide the reader. In the end, while the authors speculate about the meaning of various observations, for example about the role of the heterodimer in regulating binding of elongation factors, these ideas are not tested experimentally. While these new structures will be of interest to specialists, there are insufficient insights that will be of interest to the general readership of eLife.

Following the criticisms of the reviewer, we have re-written the manuscript that now focuses on the A49-A34.5 heterodimer exchange with the C-terminal domain of subunit A12.2 (A12.2C) as observed in the Pol I* structure. We have also added experimental data in vitro that helps understanding the dynamics of the binding of the A49-A34.5 heterodimer and A12.2C. Additionally, we have revised the figures to increase the clarity of the manuscript (for example, we have included the Pol I* and Pol I structures in Figure 1 to facilitate comparison and we included a legend indicating the colored subunits and their relationship with Pol II to guide the reader). We believe that our revised manuscript will be of interest not only to Pol I specialists, but also to the broad readership of *eLife* interested in the conformational dynamics of protein complexes and transcription. Furthermore, the manuscript highlights the power of cryo-EM to resolve distinct intermediates in biochemically homogenous samples.

Specific points:The interpretation of additional density (Figure 1—figure supplement 3) that "connects to the DML toward the A12.2 linker,” etc.) is not at all convincing or even clear as described. It is difficult to tell where in the structure the density is located due to poor labeling. Moreover, the legend states that this density is only visible at low contour levels. Without further clarification of which residues this might correspond to, this should be omitted.

As suggested by the reviewer, we have now omitted this figure and the corresponding discussion.

Reviewer #2:[…] There are some significant challenges, though, that can be addressed and would make the story seem better supported.1) The authors have a robust understanding of the three polymerases, but their comparison of similarities could use more clarification for the non-expert reader.

We have expanded the Introduction to include more background information about the Pol I structure compared to Pol II and Pol III. We also have modified figures and text to improve readability and clarity for non-expert users.

2) This is a feature of structural biology, but this manuscript seems to make substantial mechanistic conclusions from the structural models. This was particularly evident in the section describing the motions in the +1 and +2 positions. Throughout, there should be more care taken to indicate that these are hypotheses.

In the revised version of the manuscript, we have greatly reduced the section describing the motions of the +1 and +2 nucleotides, and only briefly mention these features of the active site of both Pol I and Pol I*. Throughout the revised manuscript, we now avoid making substantial conclusions from the structural models, but rather put them into context of the known literature.

3) Does the heterodimer really leave the enzyme during transcription? The manuscript draws on some literature, but is this a well-supported view? Can there be tests?

To our knowledge, there is no conclusive in vivo evidence that shows that the heterodimer stays or leaves the complex during transcription. The current view is that the heterodimer functions as a built-in general transcription factor with roles in transcription initiation and/or (early) elongation. For the PAF53/PAF49 there is also solid in vivo evidence that the interaction with Pol I is regulated and that starvation leads to the dissociation of the heterodimer from Pol I (Penrod et al., 2015). However, this does not prove that the heterodimer binds and dissociates during the transcription cycle. in vitro, the A49-A34.5 heterodimer is required for promoter-dependent transcription initiation, although the C-terminal A49 tWH domain is sufficient to restore this activity (Pilsl et al., 2016). We have now included additional experimental data that also show that the heterodimer binds with high affinity to Pol I in vitro, and cannot be displaced by the isolated A12.2C or by GMPCPP.

4) A commentary on dynamics might be interesting. The discussion of these subunits being exclusive, then swapping with transcription factors (Discussion section) would benefit from a consideration of dynamics in solution (or in living cells).

We have now included some in vitroexperimental evidence on the dynamics of heterodimer binding to Pol I by fluorescence polarization assays. Whereas we can specifically track the binding and release of the heterodimer from the Pol I core, it couldn’t be displaced from the complex by either free A12.2C or GMPCPP (Figure 3). Although we could not identify conditions (or factors) that allow dissociation of the heterodimer, the heterodimer has to dissociate from the Pol I core to allow A12.2C binding to the Rpb9-like site.

Reviewer #3:[…] I strongly support publication, upon minor textual revisions and clarification of few minor issues1) Can the author monitor "ejection" of the heterodimer upon incubation with GMPCC? This would be nice to see. This could probably be done on immobilised DNA beads and the composition of Pol I monitored over time in presence of GMPCC.

Taking into consideration the suggestions of this reviewer and reviewer #2 (see above), we used a fluorescence polarization assay to monitor binding and dissociation of the heterodimer from the Pol I core. Whereas this set up allowed us to track the specific binding of the heterodimer (Figure 3 and Figure 3—figure supplement 1), we could not observe any significant effect of GMPCPP or A12.2C on heterodimer dissociation from the Pol I core. Thus, based on these results, it is likely that the high proportion of Pol I* observed in our cryo-EM data set is due to other factors occurring during grid preparation such as a change in the salt concentration on the grid or eventually also the use of carbon-coated grids, which shifted the proportion of Pol I/Pol I* normally seen in solution. Whereas in vivo, we hypothesize that binding of elongation factors such as Spt4/Spt5 of Paf1 could expel the heterodimer.

2) Results section “Cryo-EM structures of the GMPCPP-bound Pol I elongation complex (EC): this is then the authors elaborate on how this apo PolI* is formed? And why wasn't this observed in previous reconstruction of elongating Polymerase I? Could it be that Pol I* is a less stable elongation complex and partially loses the DNA scaffold after ejection of the heterodimer? I think that an in vitro assay (and possibly re-titrating in recombinant heterodimer) would add a lot to the manuscript and help understand the mechanism.

The presence of apo Pol I* particles in this data set might parallel the proportion of Pol I not bound to DNA at this scaffold concentration (2-fold molar excess). A 2-fold molar excess of scaffold was also used by Neyer et al., who observed about 50% of the particles in an apo Pol I state (Neyer et al., Nature, 2016). We could also observe apo 14-subunit Pol I in our dataset, but did not analyze it further. In our previous data set (Tafur et al., Mol. Cell, 2016) we used a higher scaffold concentration (5-fold molar excess) presumably resulting in fewer apo 14-subunit Pol I particles. The lack of Pol I* particles observed in the previous data might result from the much lower total number of particles available for classification or from different experimental conditions (different salt concentration during grid preparation etc.). Nevertheless, we also observed Pol I classes lacking the heterodimer in the previous reconstruction of elongating Pol I (Tafur et al., Mol. Cell 2016), but no unassigned additional densities were observed and thus these particles were discarded.

3) The first paragraph of the Introduction is difficult to follow maybe rewrite more concisely?

The Introduction has been rewritten.

4) In the Accession numbers section, are the complexes named properly? Which one is the apo PolI*?

We thank the reviewer for pointing out that apo Pol I* had been not included. We now include all models and corresponding PDB/EMDB codes in the manuscript.

5) Last sentence of the Discussion: the fact the general catalytic mechanism in RNA Polymerase is conserved was already clear. I would suggest that the molecular determinants/regions involved in catalysis performs similar functions and this is now successfully proven.

The reviewer is right and in the revised version of the manuscript this sentence has been deleted.

[Editors' note: the author responses to the re-review follow.]

Essential revisions:1) The binding curves and analysis are problematic. According to the methods section, labeled heterodimer was present at 100 nM concentration in the binding studies. If this was indeed the concentration, the equation for the binding isotherm is not valid, as this approximation assumes that the labeled species is present at much lower concentration than Pol I. Either the binding data should be re-analyzed with a quadratic binding equation or the experiments should be repeated at much lower labelled heterodimer concentration.

We agree with the reviewer that analyzing our binding data is problematic given that heterodimer and Pol I were present at similar concentrations (100 nM heterodimer, 0 to 100 nM Pol I*). Following the suggestion of the reviewer, we tried to fit the data using a quadratic binding equation, but have not be able to converge to a reliable solution suggesting that the binding mode is indeed more complex than we thought. As we have no further information on the number of binding sites and the sequential binding events, we want to refrain from applying a more complex binding model. Alternatively, we have also tried to reduce the heterodimer concentration in the fluorescence polarization experiments. However, reducing the heterodimer concentration led to a very faint fluorescent signal with poor signal-to-noise.

Because the primary objective of our binding experiments has been to compare binding among the different complexes, we therefore suggest to no longer report K_D_ and Hill coefficient. Instead, we just show in Figure 3B the normalized change of anisotropy upon binding of the heterodimer to Pol I* or to wild type Pol I at a given concentration without trying to deduce any quantitative binding parameters. In addition, we also show the inability to displace the heterodimer by an excess of A12.2C or GMPCPP.

In the revised version of the manuscript, the paragraph describing the results in Figure 3B now states: “To test these hypotheses, we performed a series of fluorescence anisotropy experiments, using recombinant heterodimer, where a cysteine has been introduced in the A49 linker region for labelling with Alexa Fluor 594, and endogenously purified Pol I* (Pilsl et al., 2016) incubated with DNA (Pol I * EC) (Figure 3A). […] Because a 1:1 binding model did not allow fitting the data no attempt was made to introduce more complex binding models.”

2) The proposal that sequential binding of the N- and C-terminal domains of the heterodimer cannot explain apparent cooperativity, as the experiment only monitors binding of single monomers. It is possible that the analysis of the data as suggested above, a more complex model or a repeat of the experiment under different conditions could sort this out.

As outlined in our response to point 1), we have removed the K_D_, Hill coefficient and no longer discuss the cooperativity of binding in the text.Accordingly, we have deleted in the revised version of the manuscript our statement: “Such behavior can be explained by the sequential binding of the N-terminal and C-terminal domains where anchoring of the A49-A34.5 heterodimer to Pol I by the N-terminal dimerization domains promotes the binding of the C-terminal A49 tWH domain (Figure 3B).”

3) There was not enough information given to evaluate the competition/exchange experiment. If the heterodimer indeed binds with a 14 nM K_D_ and the complex with Pol I was present at 100 nM concentration, almost all of the proteins would be in a complex. Exchange with A12.2C would depend upon the off rate of A49-A34.5 and the on rate for A 12.2C. The complex was incubated with A12.2C for 30 minutes. Were longer time points measured? What is the K_D_ of A12.2C for Pol I? Might there be different conditions under which dimer exchange would be observed? The ultimate conclusion, that the heterodimer must first dissociate in order for A12.2C to bind is surely correct in light of the structure, but it is important that the paper has binding data that show this. A good control for the experiment would be exchange with unlabeled heterodimer, as this would provide a good benchmark. Similarly, the authors should consider repeating the GMCPP experiment at lower concentration of Pol I* – heterodimer complex, as it is possible the complex would dissociate if the concentrations were not so far above the (apparent) K_D_.

We performed the competition assay to test whether excess of A12.2C could compete for the heterodimer binding site, which we saw not to be true. As proposed by the reviewer we also performed the incubation overnight and did not see a difference. As these experiments were only performed once, we did not include them into Figure 3C, but now mention these results without showing the data. We agree with the reviewer that it would be desirable obtaining binding data for A12.2C. However, A12.2C is only a small domain (M_r_ = 10 kDa) that presumably will not easily displace the bound heterodimer. Obtaining reliable binding data for A12.2C to the Pol I* core is experimentally challenging and in our opinion beyond the scope of this manuscript. In the revised we now state: “Incubation of the sample Pol I*/A49-A34.5 sample with recombinant A12.2C (residues 79 to 125) for 30 min and overnight (data not shown) did not reduce the anisotropy (indicating the release of the heterodimer from Pol I) even at 50-fold molar excess (Figure 3C).”